# Identifying the Predictors of Community Acceptance of Waste Incineration Plants in Urban China: A Qualitative Analysis from a Public Perspective

**DOI:** 10.3390/ijerph181910189

**Published:** 2021-09-28

**Authors:** Yanbo Zhang, Yong Liu, Keyu Zhai

**Affiliations:** 1School of Management Engineering & Business, Hebei University of Engineering, Handan 056038, China; zhangyanbo0530@hebeu.edu.cn; 2Fine Arts College, Shanghai University, Shanghai 200444, China; 3School of Foreign Studies, China University of Mining and Technology, Xuzhou 221116, China; Keyu.zhai@cumt.edu.cn

**Keywords:** community acceptance, waste incineration plant, the public, determinants, China

## Abstract

Due to concerns about consequences to public health, the ecosystem, the natural landscape etc., the planning and construction of waste incineration plants always gives rise to a reaction and even protests from local communities. This study aims to investigate the determinants affecting public acceptance of waste incinerators. We contribute to the existing knowledge in the following ways: (1) this study undertook a qualitative analysis on community acceptance of nimby facilities in the context of China for the first time; (2) through qualitative interview analysis, we emphasize the impact of interactions among multiple factors regarding the acceptance of waste incinerators; (3) we finally construct a framework to systematically explain the formation mechanism of community acceptance of waste incineration plants. Employing in-depth interviews with 22 representative residents, the results indicate that from the perspective of externality, risk perception has a significant negative impact, whereas the effects of benefit perception are positive. In terms of interaction between government and citizen, both justice perception and political efficacy are positive. Social situational factors positively promote community acceptance. Lastly, the impact of individual cognition is mixed. This study has the potential to make a significant difference in better community governance and environment-friendly cities.

## 1. Introduction

The rapid expansion of urban China and the upsurge of migrant workers has intensified the risk of urban China becoming besieged by waste. With the aim of solving this issue, incineration has been on the government agenda due to its advantages of converting waste to energy and alleviating land scarcity pressure [1]. By 2019, China had 389 waste incinerators in operation, whose treatment capacity accounted for 57.70% of municipal solid waste (MSW) [2]. Incineration is becoming the dominant solution to deal with MSW in China.

The expansion of waste incineration capacity is necessary to lessen the heightening ‘waste crisis’ of Chinese cities, and is gaining growing support among the general population. However, the planning and construction of waste incineration plants always brings about reactions and even protests among local communities, who are concerned about negative impacts on public health, the ecosystem and the natural landscape [3,4,5].The gap between general support and local opposition to such facilities has attracted increasing attention from scholars, as well as from government agencies and the incineration industry. The gap is indeed a complex issue, with political, economic and social implications. This directly influences the extent of public acceptance articulated by residents living in the vicinity, which is of critical significance for the successful siting of waste incinerators in any society [6,7,8,9].

The current literature on acceptance to waste incinerators is almost exclusively based on semi-structured questionnaires. Existing studies have confirmed that distance, perception of need, risk perception, benefit perception, social trust, distributional equity, procedural justice and place attachment are positively related to public attitude to waste-to-energy plants [10,11,12,13]. These works try to address the influencing factors of public acceptance to waste incinerators with assumed hypotheses which may neglect factors with greater explanatory power regarding local residents’ attitudes, and which fail to adequately grasp the complexity of residents’ motives. In addition, these factors only partially appear in certain research, and a complete system framework regarding public acceptance has not been put forward. In this case, the factors of the same root are divided into separate individuals. This means that with regard to environmental governance, governments are inclined to pay attention to separate factors while ignoring the importance of a complete governance system framework. Therefore, in many cases, the absence of a deep understanding of various determinants of public acceptance and the relationship between them cannot well inform the good environmental governance. Without a complete framework of influencing factors, the government seems to have lost its sense of the relationships among the various factors. Furthermore, the lack of a complete system framework will make the governments unconsciously fall into a state of passive governance. In this state, only when problems arise will governments passively solve them, often in a scattered and unsystematic way. When there is no problem, governments do not know how to improve public acceptance of environmental governance. The final result is the overall decline of the government’s environmental governance capacity and an increase of public discontent. Thus, the question of whether there is a systematic and complete factor framework with which to explain the differences in public acceptance is very important.

Indeed, successive anti-incinerator campaigns in China have attracted the attention of scholars, even though their efforts have mainly applied social movement approaches. Some researchers have argued that digital time empowers activists with opportunities to take strategic actions to mobilize or legitimize their resistance [14,15]. Another group of scholars noted the policy changes resulting from local protests, highlighting the significance of policy initiatives advocated by rational actors in a restrictive political setting [16,17]. The results from the studies aforementioned underemphasized the human motivations in shaping the formation of public attitudes and their possible behavior, and cannot effectively inform policy development and risk communication regarding waste incinerator siting.

Based on the growing body of research regarding public attitudes to the siting of renewable energies [18,19,20], this paper aims to obtain a better understanding of public acceptance toward waste incinerators in China. More specifically, we seek to identify individual and social factors which are relevant to public acceptance. To this end, we conducted in-depth interviews with 22 representative residents living in the vicinity of waste incinerators in different regions in China. This study enriches the current literature on public acceptance to controversial facilities in an authoritarian setting. The analysis with qualitative design provides a systemic understanding of the factors and emotions which shape public acceptance to waste incinerators, and could be utilized to select the effective measures to enhance public acceptance to such facilities. Opaque environmental information leads to a decline in public trust of the government. The lack of independence of public participation procedures leads to public doubts about the consequences of environmental projects. In the process of evaluating environmental project, contradictions are often prominent. Therefore, it is necessary to solve environmental problems through interactions with the public. Moreover, there are shared interests between government evaluation organizations and organizations responsible for constructing environmental projects. Therefore, the government’s environmental assessments are often not objective enough to fully consider the interests of the public. Thus, we discuss this issue from the perspective of the general public. This perspective can protect the environmental rights of vulnerable groups and prevent damage, perpetrated by strong groups upon vulnerable groups. Meanwhile, this perspective can promote the implementation of democratic decision-making and provide a channel which reflects public opinion. Consequently, our study could make a great contribution to social justice.

The remainder of this paper is organized as follows. Section 2 offers a literature review, which helps us to understand why the NIMBY protests occurred, and the effects and leadership of NIMBY resistance. Section 3 introduces the methodology and data obtained in the present study. Section 4 shows our results and discussion. Conclusions and some policy implications are presented in Section 5.

## 2. Literature Review

### 2.1. Why did NIMBY Protests Occur?

In order to put forward more targeted regulatory strategies, the current study attempts to deconstruct the logic of NIMBY resistance from the perspectives of social psychology and social governance. NIMBY conflict is not only closely related to public risk perception, but also the result of a variety of psychological factors in a specific situation [21]. The essence of perceived risk to NIMBY facility risk is knowledge production, and the differentiated cognition of risk knowledge among different subjects is an important component of NIMBY conflict [22]. At the micro level, Anderson & Schirmer analyzes how Canberra residents’ resistance to gas-fired power stations occurs, and the result shows that local attachment is the catalyst to stimulate residents’ resistance to the struggle [23]. Some studies suggest that NIMBY resistance is the result of the dynamic interaction of a series of social and psychological factors. Based on the generation process of organizational behavior “need-motivation-behavior”, a study argues the emergence and development of NIMBY resistance is closely related to cognition, expectation, emotion, rational consideration, need and motivation caused by NIMBY plot, so NIMBY resistance is a complex process of “cognition-query-value appeal-motivation-collective action” [24]. The driving force for the continuous development of NIMBY resistance is neither the contradiction intensified by the unfair distribution of economic interests, nor the protest of environmental rights defenders against the dominance of administrative power, but the interactive force of the complex chain composed of “value, rationality and power” [25]. Meanwhile, a series of social psychological analysis models are constructed to explain the occurrence of NIMBY resistance. For example, Wang et al. believe that “differential trust pattern” is the core factor that leads to the resistance of Chinese neighbors, and constructs a three-dimensional interpretation framework including identity characteristics, risk opportunities and media construction [26]. Hou et al. constructed an integrated analysis model of the evolution of NIMBY resistance, in which the participants’ risk perception, frustration and distrust are considered as important internal driving forces [27].

Another major perspective is social governance. The political opportunity structure mainly focuses on the study of the political environment in which the social movement is located and the external resources available to the movement participants, and emphasizes how the changes of laws and policies, political system and elite sympathy affect the action strategy, resistance potential, resistance frequency and organization formation of the protesters [21,25,26]. Thus, it emphasizes the interaction between participants and political subjects. From the perspective of political opportunity structure, the imbalance between government supply capacity and public participation demands leads to the “relative closeness” of the government, the division of elite groups and the construction of external alliances, and the low tolerance of the government. However, Gao found that the structure of political opportunity is only an important external condition for the occurrence of NIMBY resistance, and the interaction between political opportunity structure, social network of actors and the use of the Internet, as well as fermenting and mobilizing constitutes the chain of NIMBY resistance [28]. Similarly, Bu believed that the structure of political opportunity only emphasizes the analysis of macro structure and institutional level, and the occurrence of NIMBY resistance also needs to create a certain space for discourse resistance [29]. Wright & Boudet analyzed 20 communities involved in 18 energy NIMBY facilities in California, and the results showed that political opportunity structure and resource endowment were not the necessary conditions for NIMBY resistance. Community situation could better induce NIMBY resistance by shaping resistance motivation and combining motivation with ability [30]. In addition, based on the bonding capital and bridging capital of social capital, some studies described in detail how the catalysis, formation, maintenance and diffusion of the protest social network develop and evolve through different forms of social capital [31]. Figure 1 indicates the occurrence framework of NIMBY resistance. 

### 2.2. Effects of NIMBY Resistance

NIMBY resistance will lead to policy changes and social innovation. Regarding conventional policy changes, due to the positive response of local governments to residents’ demands, NIMBY resistance could lead to the establishment of a new public consultation mechanism [16]. Accordingly, changes of unconventional policy are also an important consequence of NIMBY resistance. The openness of the political system and the consensus of policy change in the political system are important factors affecting policy changes regarding NIMBY facility location [32]. In addition, policy initiatives by NGOs, public figures or activists among the general public are the necessary conditions to connect resistance and policy change [17]. In the Chinese context, the unconventional policy changes caused by urban NIMBY resistance may be attributed to the technocratic governance mode of public policy, that is, public decision-making relies too much on technical experts, which means that policy-making falls into the technical Leviathan dilemma [33].

Also, some studies have analyzed the impact of NIMBY resistance on social innovation; NIMBY resistance is not short-sighted and narrow-minded egoism, but rather, promotes social innovation and development. NIMBY’s “beauty” is mainly reflected in three areas: political innovation, such as the birth of new social networks, as well as institutional and policy changes; scientific and technological innovation, such as energy grid improvement, new nuclear waste treatment technology; and social innovations, such as the emergence of new NGOs [34]. In the process of NIMBY resistance, the public is constantly engaging in social learning, which is reflected in the collective reflection on the environmental problems embodied by NIMBY facilities. Meanwhile, the public tries to establish a communication mechanism with the local government to reach a solution to these issues [35]. In addition, the emergence of NGOs is also the embodiment of social innovation. NIMBY resistance constructs an environmental civil society through four closed-loop processes: movement space separation, environmental citizen production, environmental NGO formation, and environmental civil society operation and retreat [36]. Meanwhile, regarding government learning, the task orientation of superior government, the problem orientation of crisis learning results and the demand orientation of multiple stakeholders outside the government are the driving factors of local government crisis learning. The guarantee of superior government resources and the flow of organizational memory within the government are the necessary conditions for local government crisis learning [37].

### 2.3. Governance of NIMBY Resistance

Most studies deconstruct NIMBY from the perspective of governance theory, and then put forward corresponding strategies. Some scholars believe that resistance to NIMBY is not only related to differences of interests, but is also affected by deeper value factors. The value deviation and distortion of public interest intensify contradictions, leading to resistance. Herein, it is proposed that we need to pay special attention to the public interest in NIMBY facility planning and location, and cultivate consensus public value to resolve these contradictions [38]. In addition, some scholars believe that the current government-led policy system blocks the effective participation of the public in the NIMBY facility siting policy, which seriously weakens its legitimacy and negatively affects public acceptance. Thus, fine policy design in the terms of the timing, degree and form of policy participation could effectively reduce resistance to NIMBY projects [39]. Others have identified the interactions among the different alliances which are behind resistance to NIMBY, and noted that the interaction mechanisms of these alliances are key to the resolution of such resistance. Therefore, reshaping and optimizing the governance structure of NIMBY facility location and promoting positive interactions between different alliances and entities are key to minimizing resistance [40].

Based on the theory of environmental justice, some scholars have argued that the infringement of citizens’ environmental rights and the imbalance of environmental risks and interests are the main reasons for NIMBY resistance [41,42]. Thus, they suggested that public rights and reasonable compensation should be guaranteed to realize environmental equity and ensure the justice of environmental decision-making. Also, some studies stated that a lack of consultation is the root of NIMBY resistance. Establishing the thinking of a deliberative democracy and broadening the channels of public consultation could also minimize resistance [43].

## 3. Methodology

### 3.1. In-Depth Interview Organization

The data in the study were based on in-depth interviews. The purpose of the in-depth interviews was to examine whether the public was willing to accept the construction of a waste incineration plant near their places of residence, and what factors affected their attitudes towards the construction of such a site. Compared with quantitative methods, the qualitative data obtained through interviews were comprehensive and in-depth. The collection of rich, vivid and detailed information can much better reflect the true nature of a situation [44]. Based on a literature review, interview questions were composed which involved public awareness of waste incineration and acceptability of waste incineration power plants, as well as of other local government initiatives. Different approaches were used, i.e., social psychology, social governance, social exchange theory, etc. These approaches were integrated to contextualize the interview questions and subsequently analyze the data. These approaches make it possible to study the social actions of individuals from psychological, governmental and exchange perspectives. The present study investigated the social taking into account the above dimensions. We applied these approaches to contextualize the interview questions. Regarding data analyses, in the analytical interview results, we also incorporated these approaches to inform the in-depth discussions presented in this report.

The data collection process was carried out according to the theoretical sampling procedure of the grounded theory research method. We first used the purposive sampling method to invite residents around the waste incineration plant to participate through WeChat social tools. Then, through snowball sampling, we invited interviewees to recommend residents around the same waste incineration plant in order to expand the number of potential interviewees. The selection of interviewees followed three basic conditions: first, we had to comply with the wishes of interviewees; second, the respondents had to possess a certain level of knowledge and understanding on the nearby waste incineration power plant, e.g., facility construction, environment impact assessment, government deeds and efforts; third, the number of respondents from the same waste incineration power plant could not exceed four. Finally, 22 residents were interviewed who lived in the vicinity of 11 different waste incineration power plants (Table 1).

Telephone interviews were used in this study. This choice was mainly based on two factors: first, the novel coronavirus outbreak at the end of 2019 disrupted the original plan of conducting face-to-face interviews; and second, telephone interviews diminish the concerns of the interviewees to a certain extent, create a safe interview space, ensure that the interviewees can speak freely, share real feelings and experiences, and provide richer information. With the consent of all interviewees, all interview contents were recorded, which ensured the integrity of the final presentation. In addition, the text materials provided by some interviewees and the information obtained from the Internet for specific cases provide a powerful supplement to the interview materials. The average duration of each interview was about 1.5 h. Chinese was used in the interview process; in this report, English translations of interview transcriptions were used. The final transcript was reviewed by two Chinese professors of English and one British professor of urban studies.

### 3.2. Data Analysis

This study employed thematic analyses to identify and analyze the patterns of interview data. This approach is widely used in a range of theoretical frameworks [45]. Based on Corbin and Strauss, this study selected opening coding, axial coding and selective coding to analyze interview data [46]. In the coding process, a continuous comparison method was used to refine new concepts and categories until the new interview materials no longer contributed to the theoretical construction. In order to improve the scientificity and objectivity of coding and reduce individual subjectivity biases within the coding results, we invited researchers with rich experience to code the first text together. This study used the Nvivo11 Software as an auxiliary tool for data coding; this qualitative analysis software has been widely used to analyze the public acceptance of wind power plants [47]. Through the coding process, we identified some key themes and subthemes which we could use to explain resident perceptions (Table 2).

Finally, a variety of themes were developed to deal with the main research question, including the externality of NIMBY facilities, the political behavior of the government, social situational factors, and individual cognition. Based on these codes, detailed analyses will be shown as we discuss each main theme.

## 4. Results and Analysis

### 4.1. Externality of NIMBY Facilities

Externality is induced by an individual’s diverse social activities. Therefore, any social field has the possibility of externality [48]. A waste incineration power plant is a public goods with typical externalities. The purpose of such a project is to alleviate the problem of urban waste and create a good environment for the development of the whole city and its residents. However, the welfare level of residents around the site is weakened by its presence, i.e., there are risks negative effects on human health, shrinkage of the value of real estate, psychological discomfort and damage to the ecological environment. These lead to a certain number of negative externalities.


*Fly ash from waste incineration will eventually enter the lake. In fact, the waste incineration plant will emit smoke, which will sink into the lake after being discharged into the atmosphere. Let’s not say that the residents around us, even those with a little common sense, know that this is really undesirable. Waste incineration plant is too harmful to our health and life*
*.Although dioxins emitted from waste incineration are invisible and untouchable for a long time, they can cause cancer.*
(Wu, Male, 47 years old, Self-employed people)


*The establishment of incineration plant has a great impact on the surrounding buildings. In the past, the house price of our community could easily be sold to 14,500 yuan per square meter, but now because of the construction of waste incineration plant, the price of 8000 yuan or 9000 yuan per square meter is not accepted.*
(Wand, Female, 59 years old, Engineer)

In addition, respondents believed that some compensation measures taken by local governments, such as the introduction of tap water and widening roads, were only remedial measures or strategic actions. Local residents believed that these measures had not brought about substantial benefits in the surrounding areas. As the bearers of the negative externalities of waste incineration projects, the surrounding residents are also rational “economic men”, who have the motivation and tendency to pursue their own utility maximization. Faced with the potential risks and uncertainties of living in the vicinity of a waste incineration power generation project, as a rational “economic man”, the surrounding residents will not be indifferent to the potential benefits. Potential consideration based on negative externalities, such as risk or benefit perception, affects their acceptance attitude towards waste-to-energy power plants.

### 4.2. Interactions between Government and Residents

The results show that justice perception and political efficacy are two important factors in the interactions between government and residents which affect the public acceptance. A perceived lack of justice is an important factor that causes the public to not accept the construction of waste incineration power plant.


*The local government is eager to build a waste incineration plant. In the early stage, it has not communicated with the surrounding residents well. Before the relevant information is made public, the local government starts purchasing equipment. Many owners will spontaneously think whether there is corruption in the process. We do not mean malicious speculation, but there is no other channel to give us a reasonable explanation.*
(Wang, Female, 30 years old, Ph.D. student)


*The second phase of the waste incineration plant was not built here, but in other places. But because of the strong opposition of the local people in that place, the government forced us to build a waste incineration plant here. Why? That place is a treasure land of geomancy. Aren’t we?*
(Huan, Female, 38 years old, Freelancer)

Social exchange theory holds that people in a society are essentially in a kind of exchange relationship, whereby mutual benefit is the basic criterion to drive social exchange activities. Based on social exchange theory, if the public’s rights in the public affairs of a waste incineration power plant are guaranteed and realized, and they really feel that the waste incineration power plant has safeguarded their own interests, then they will trust the local government more, support the project and actively cooperate with the relevant decisions of the local government, i.e., they will actively accept the waste incineration power plant. Existing studies have also confirmed that different dimensions of justice perception have a positive impact on the public acceptance of NIMBY facilities [49,50].

Political efficacy is an important part of the political psychological activities of the general public. Citizens’ perception and evaluations of their own political ability regarding political activities or processes are considered to be an important factor affecting individual political behavior [51]. The processes of site selection, construction and the operation of a waste incineration power plant are also political, involving interactions among multiple stakeholders, such as the government, developers and residents around the site. Residents’ attitudes and protest behaviors are also driven by a sense of political efficacy.


*I said I do not object, and I cannot object. I cannot take care of other people’s heaps at my door. Can I take care of the garbage incineration power plant several kilometers away from me? is it? So, I cannot object either.*
(Zhao, Male, 50 years old, Farmer)

Internal political efficacy shapes the attitudes of each individual, and even drives them to take action by strengthening emotional mood and weakening potential negative policy influence [52]. Individuals with a strong sense of internal political efficacy generally have high confidence that they will succeed in achieving their desired goal through participation in political activities, thereby strengthening their efforts toward the desired goal. In the face of waste incineration projects that affect people’s lives, when individuals think that they have the ability to influence the relevant political decisions, psychological cognitive resources will positively strengthen their willingness to protect their own areas, and therefore, to generate resistance to waste incineration projects. Waste incineration power plants are public utilities which are strongly connected to the interests of the surrounding residents. In the face of waste incineration power generation projects, the values of the surrounding residents are generally inconsistent with those of the local government. When a nearby resident thinks that his or her demands can be met by the local government, his or her willingness to express his or her policy attitude will be enhanced; at the same time, his or her willingness to express real policy attitude will be weakened. The prediction effect of collective political efficacy on public participation in NIMBY protest action has also been verified by existing studies [53,54,55,56]. Therefore, collective political efficacy will affect public acceptance of waste incineration power plants.

Although the various channels and forms of public participation have been continuously expanded in recent years, public participation in public affairs is still limited. In terms of sensitive projects such as waste incineration sites, the “localized/non-transparent” behavior of local governments has seriously limited the space for public participation, weakening the public’s sense of political efficacy to a certain extent.


*At that time, the incineration plant had already been built. When we stopped it, we might have limited capacity on the one hand, but on the other hand, we might not be able to overcome some political and social resources, so we did not succeed.*
(Jiang, Female, 38 years old, Salesperson)


*The communication with the government has no effect at all. The reality is that we solve this problem by only calling the mayor’s hotline and the State Council’s complaints every day. I think it is very inefficient for the surrounding residents to make repeated complaints, but it is the only channel for us to express our wishes.*
(Wang, Female, 30 years old, Ph.D. student)

Due to the unfavorable position of political efficacy, the public cannot significantly influence the decision-making of the local government. They can only express their inner resentment and dissatisfaction at being forced to adapt to living with a waste incineration power plant in the vicinity of their home, or they can resist. Thus, political efficacy has a significant positive impact on public acceptance of waste incineration power plants.

### 4.3. Social Situational Factor

Our results show that the major social factor affecting public acceptance of waste incineration power plants is public trust. In this study, the objects of public trust are local government agencies, government staff, operating agencies and staff closely related to the plant itself. Respondents believe that local governments have insufficient capacity to address the problems of waste-to-energy power plants.


*Every week, the district government held a meeting with the residents to talk with us about how to solve the problem, but later found that there was almost no way. We had a time with local governments to solve the problem before, but the problem was not solved after the time, so we didn’t know whether the local government could solve the problem well.*
(Fei, Female, 40 years old, Salesperson)


*In addition, the local government did not protect the interests of local residents. The operation subject of waste incineration power plant only focuses on private interests, and will not fulfill social responsibilities.*
(Wang, Female, 30 years old, Ph.D. student)


*We first asked the mayor of jiangxintun town in Xianghe for help, but from his attitude, we felt that he didn’t want to take care of it. He always refuses. Second, he is also defending the big factories, rather than fighting for the interests of local residents.*
(Chang, Male, 57 years old, Designer)


*A public enterprise as big as this cannot be operated by a private enterprise, because the private enterprise only pays attention to its own interests, so it will not care for the local people.*
(Su, Male, 33 years old, Cybercafe administrator)

The operation of a waste incineration power plant is not standardized, which has a negative impact on the surrounding residents, leading to a loss of public trust and strengthening public opposition to such projects.


*The first phase project of the incineration plant really has a great pungent smell, and the garbage stacking of the incineration plant is harmful to human body. We can’t blame the surrounding residents and villagers. The pollution of waste incineration plant has happened for a long time. Why doesn’t he object?*
(Zhang, Female, 63 years old, Retired)

Social capital theory holds that people will form a continuous social network through interactions and contact with others, and that this network can encourage actors to obtain resources for their own development. Trust is an important element in this network. In the context of social risk, the positive role of trust is more prominent, because its existence can overcome the lack of social rationality and simplify the complexity of the risk situation [57]. In addition, by squeezing or replacing external uncertainty with an internal certainty, trust can weaken the panic and negative emotions caused by uncertainty in social relations [58]. Social actors with more trust resources can positively intensify adhesion with other social members, and can also strengthen the recognition and understanding of other social members by simplifying complexities and uncertainties, thereby increasing support and cooperation. Therefore, public trust plays an important role in explaining public acceptance of waste incineration power plants.

### 4.4. Individual Cognition

Place attachment is used to describe the emotional attachment or bond between people and a specific place. It is an important concept with which to measure the relationship between people and the physical environment or a geographical space [59]. Individuals with strong emotional attachments to a place tend to show higher interest in the public affairs of that place, and are more willing to participate in the affairs thereof [60]. Regarding individuals residing in the vicinity of a waste incineration power plant, place attachment refers to the emotional connection between the public and the physical environment or geographical space of their residence, which is an important emotional factor affecting their attitude towards the waste incineration power plant. Due to the potential risk of damage to the living environment, the emergence of waste-to-energy power plants has stimulated the willingness of residents to protect the living environment and maintain their own quality of life, which brings about psychological resistance to such projects.


*In fact, even if I was asked to move, I would not accept it. The place where I live is beautiful and the water I drink is natural water. There is also an AAAA scenic spot here—Heimilu peak forest park. It is a natural oxygen bar. The air quality is very good, and its ecological value is priceless. The garbage incineration plant is built here, just like a tumor on a beautiful woman’s waist. As time goes on, if she is purulent again, she will die in the end.*
(Liu, Female, 40 years old, Volunteer)

In addition, the risk characteristics of waste incineration power generation projects have significant scientific and technological attributes. The public’s cognition and attitude are closely related to their familiarity and scientific literacy with waste incineration technology and processes. Due to a lack of relevant knowledge, the public is often hostile to new technologies [61]. In risk communication activities, local governments and technical experts should focus on the public’s characteristics and carry out targeted science popularization so as to enhance the scientific and technological knowledge.

*After the completion of the waste incineration plant, the first and most important impact is that dioxin, which is internationally recognized as a class I carcinogen, will be discharged during the waste incineration, and the emission of dioxin cannot be monitored or controlled. This is our first concern*.(Zhou, Female, 69 years old, Retire teacher)

Overall, the sample in this study did not have extensive knowledge of waste incineration; as such, it is very important to promote the spread of knowledge and risk communication [62]. Moreover, popular science education is an important factor capable of enhancing the public acceptance [63], and higher education development is closely connected to regional inequality and public consciousness [64].

## 5. Conclusions

The resistance of local residents to NIMBY facilities based upon their the location and construction profoundly reflects the complexity of governance of social problems in the context of risk society, and reveals the short-sighted behavior of public policy-makers and the failure of local governments to deal with thorny social problems. Incineration plants are a common example of NIMBY facilities that are resisted by the public. Therefore, our study contributes to the existing literature on the multifaceted determinants of public acceptance of NIMBY facilities. Using interview data from urban China, our work identifies the determinants of public acceptance of waste incineration, and describes their influential mechanism. Based on the analysis, this study constructs a systematic framework to explain the mechanism of public acceptance of waste incinerators, and will benefit future urban governance and academic studies, because residents tend to exhibit varying degrees of acceptance to NIMBY facilities in different communities. Figure 2 depicts the theoretical framework of community acceptance of waste incineration plants. Compared with previous studies, our framework does not explicitly include distance and perception of need. This is because the interviewees we selected were all residents near the waste incineration plant, and distance was therefore an unchangeable factor for them. 

Our study focuses on how to improve the public acceptance of individuals living in the vicinity of a planned waste incineration plant. Distance is a significant but unchangeable factor, and as such, was not included in our framework. Regarding the perception of need, noncore areas with simple social structures and low population mobility are most commonly selected. Thus, there is little difference in residents’ perception of need, and our empirical results did not reflect differences and/or an influence of this factor. Figure 2 presents a systematic framework of influencing factors on public acceptance in the field of environmental governance. This framework is not only of great significance for systematic academic research, but may also play a guiding role for practical governance.

Regarding the externality of NIMBY facilities, our results suggest that risk perception has a significant negative impact on public acceptance, whereas the effects of benefit perception are positive. The negative impact of public perceived risk regarding the acceptability of waste incineration power plants is more deeply reflected in the negative externality of waste incineration power plants. Due to concerns and the subjective construction of potential risks by the general public, a relatively strong perceived risk is observed, which drives the public to resist the construction of such a project near their homes. According to the interview data, this risk perception is mainly reflected in the negative impact of waste incineration power plants on residents’ health, psychology and the ecological environment. The results of benefit perception show that when analyzing the public acceptability of NIMBY facilities, we need to pay more attention to the different effects of the various dimensions of benefit perception on public acceptability.

In terms of interactions between governments and citizens, perceived justice and political efficacy are viewed as positive. The results show that high justice perception leads to higher acceptability, which is consistent with existing research findings [49]. In policy areas that are more likely to cause disputes, especially in the field of NIMBY facility location, local governments generally tend to operate policies through a “localized/non-transparent” approach. In this kind of social governance, public participation has strong limitations, i.e., it is difficult for the public to interact with the local government. In a situation of no interaction, it is difficult for the public to have a positive justice experience. Regarding waste incineration power plants, low political efficacy makes residents become “environmental refugees”. Although residents will take institutionalized or noninstitutionalized measures to prevent the construction of waste incineration power generation projects near their homes, they are often unable achieve the expected action effect, and so are forced to accept the existence of the facilities, albeit with resignation. In addition, the development of an information society has had an important impact on the political opportunity structure and the political efficacy of waste incineration plants; social movement activists in an authoritative society can use the Internet to influence the political efficacy of the government. Information technology has the potential to promote resistance and affect the political opportunity structure of a country or region. Social movements based on the Internet are often able to obtain widespread social support. Therefore, the public will obtain more positive externalities of the political opportunity structure, greatly enhancing the political efficacy of governments. The information communication technology represented by the Internet fundamentally challenges the control of the government. The government can no longer exercise political power at will; this will reduce the political efficacy of government repression to some extent. Thus, political efficacy is the result of the political opportunity structure.

Based on the results of interviews, public trust has a significant and positive impact on the public acceptance of waste incineration power plants. Public trust in the interviewees’ narratives was mainly reflected in two aspects: intention-based trust and ability-based trust. Local governments and operators of waste-to-energy power plants cannot win people’s trust through specific practices which weaken public acceptance of waste-to-energy power plants.

Place attachment has a significant and negative impact on public acceptance of waste incineration power plants. Our results provide a new perspective on the NIMBY phenomenon, that is, public opposition to the idea that waste incineration power plants should be built in their own backyards is a kind of local protection mentality. The public demonstrates emotional dependence on the physical environment around their residence, giving rise to concerns that power plants will cause potential damage to their residence environment, thus enhancing resistance to the construction of such facilities. In addition, the public has limited knowledge about waste incineration through self-learning, and attitudes toward waste incineration power generation projects will affect the intake of waste incineration information to a certain extent. As a consequence, the public’s understanding of waste incineration may remain limited. Knowledge about the environment is not only effective when risk is encountered, but should be reflected in education and learning. Therefore, knowledge is more reflected in individual cognition, because no one has any right or power to force others to learn.

Public acceptance of NIMBY facilities is the result of a combination of macro-institutional arrangements and microfactors. Governmental strategies for the public acceptance of NIMBY facilities not only need to discuss the design of accurate policy tools from the microlevel to shape public attitudes, but also need to deeply analyze how to enhance governance ability and the legitimacy of NIMBY risk by strengthening the system construction on a macrolevel. On the macrolevel, the improvement of public acceptance requires continuous improvement of the relevant institutional arrangements. On the microlevel, the improvement of the public acceptance requires the comprehensive use of diversified and accurate policy tools.

The acceptance by the general public of the construction of garbage dumps requires interactions among multiple stakeholders, e.g., the government, the construction company, residents and NGOs etc. Generally speaking, without the intervention of other stakeholders, residents tend to oppose the construction of waste incineration plants near their residence. Therefore, future studies should explore how to balance the interests of all parties from multiple angles, rather than just improving legitimacy according to public feelings.

## Figures and Tables

**Figure 1 ijerph-18-10189-f001:**
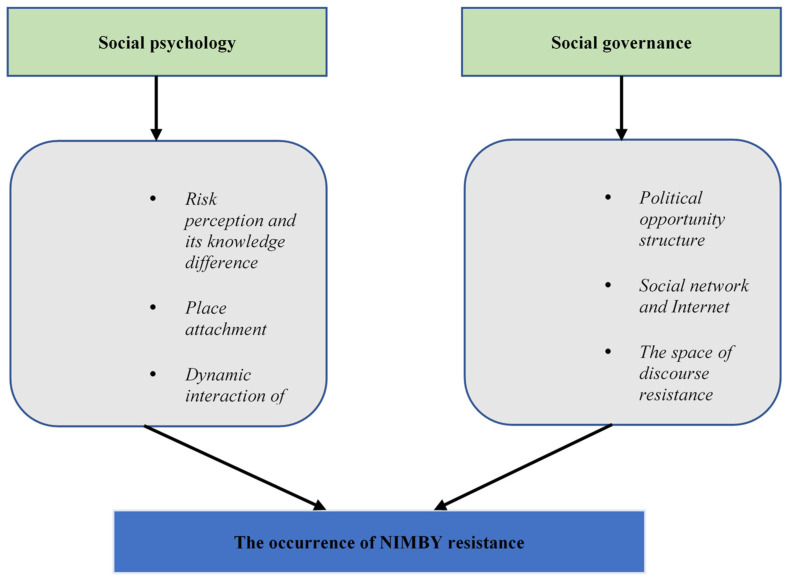
Occurrence framework of NIMBY resistance.

**Figure 2 ijerph-18-10189-f002:**
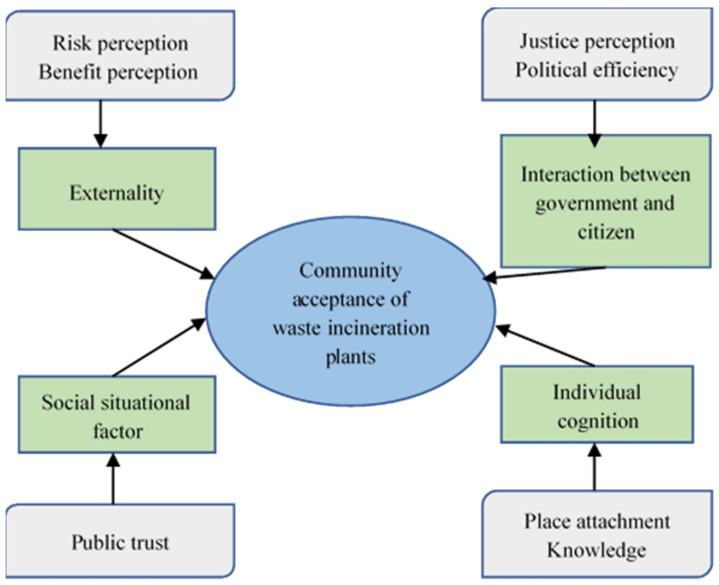
Theoretical framework of community acceptance of waste incineration plants.

**Table 1 ijerph-18-10189-t001:** Interviewees and WtE Power Plant.

Interviewees	WtE Power Plants	Location	Operate Since	Protest or Not
RO1/R06/R09/R22	Nanlang Garbage Treatment Base	Zhongshan	2014	No
R02	Dagongcun WIP	Beijing	2015	Yes
R03	Qiandaohu WIP	Hangzhou	2018	Yes
R04/R08/R21	Heimifeng WIP	Changsha	2018	Yes
R05	Jiujiang WIP	Jiujiang	Relocated	Yes
R07	Mayong WIP	Dongguan	2017	Yes
R10/R12/R13	Chengmai WIP III	Haikou	2020	Yes
R11	Taoyuan WIP	Changde	Under Construction	Yes
R14	Tengao WIP	Anshan	2020	Yes
R15/R16/R17/R20	Qionghai WIP	Qionghai extension	2021	Yes
R18/R19	Dachang WIP	Langfang	Cancelled	Yes

**Table 2 ijerph-18-10189-t002:** Coding results.

Themes and Sub Themes	Frequency
**Public Scrutiny**	5
**the significance of waste incineration plants**	7
**past similar experience**	10
**distance to waste incineration plants**	26
**governments’ slow response to the public**	9
**family structure**	3
**knowledge on waste incineration plants**	20
sort garbage	18
**health**	11
rhinitis	4
pungent odor	16
toxic gas (dioxin)	11
**the action of the public**	15
petition	3
**receptive attitudes**	15
accept with conditions	6
**interpersonal justice**	21
pressure and threaten	6
inspect	7
encourage	2
advertise	3
**pollution treatment**	9
water	8
soil	2
air	5
**perceived emotional risk**	4
**perceived interests**	3
devalue of housing	12
devalue of investment environment	7
loss of talents	3
bad impacts on local development	8
**political efficiency**	0
external	11
internal	12
**public trust**	4
distrust to government	12
distrust to enterprise	27
**information justice**	14
governments’ refuting to public emission information	4
Public information of EIA solicitation is hidden	2
**justice of the procedure**	30
lack of representativeness of the participants of EIA	7
the public does not receive project information	5
**Justice of the result**	9
harms to residents’ health	3
**education level**	2
**reasonability of site selection**	12
lack of reasonability of city planning	12
lack of reasonability of site selection	9
**consciousness of environmental protection**	4
**age**	3

The bold means they are main themes and the others are sub-themes following the main themes.

## Data Availability

The data presented in this study are available on request from the first author.

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
