# Peer review of "Identifying the Predictors of Community Acceptance of Waste Incineration Plants in Urban China: A Qualitative Analysis from a Public Perspective"

_ijerph, 2021, doi:10.3390/ijerph181910189_

Round 1

Reviewer 1 Report

As noted by the authors in the introduction to this paper, understanding community acceptance of waste incineration plants in China is of critical importance because of China’s huge investment in incineration as a current and future energy source. Hence, the issue is a significant one, but I had a number of concerns about the paper that I list below.

First, the contribution of the article is not clear. The introduction notes that “Existing studies have confirmed that distance, perception of need, risk perception, benefit perception, social trust, distributional equity, procedural justice and place attachment are positively related to public attitude to waste-to-energy plants.” The conclusion of the article presents a theoretical framework of community acceptance of waste incineration plants, based on the interview findings, that consists of risk/benefit perception, public trust, justice perception, political efficacy, place attachment and knowledge. Assuming that justice perception encompasses distributional equity and procedural justice, and that social trust is equivalent to public trust, the framework adds only knowledge and political efficacy. However, the term knowledge is used in the paper to describe knowledge of the risks of incineration (e.g. l. 88, l.410-416), so it seems to be part of the risk perception element rather than something new. Distance and perception of need are absent from the framework.

With regards to political efficacy, there is confusion between the terms political efficacy and political efficiency (e.g. l. 311-327, Figure 2). I believe that the intention is to use the term political efficacy throughout.

The literature review and Figure 1 talk about political opportunity structure as a key element in NIMBY resistance, but this concept is not mentioned elsewhere in the paper nor in the final theoretical framework. There is no discussion of the link between political opportunity structure and political efficacy.

The decision to use 11 different WtE (waste-to-energy) power plants and only one or two interviews from each site makes it very difficult for the reader to understand the context of opposition presented. No background is provided on any of the WtE facilities, such as how long they have been operating, the level of opposition that they faced prior to approval and during operation, any record of negative environmental impacts, or any benefits provided to the community.

It is not clear how participants were selected for interviews. Participants around each WtE site were recruited by means of social media and snowballing, but only one or two participants were eventually selected per site. How many individuals expressed an interest in participating in the study at each site and what criteria were used to choose those who would participate? The authors state that participants were chosen who ‘have a clear understanding on the nearby waste incineration power plant’. How was this understanding determined? Without more details, the selection of participants appears arbitrary.

Finally, for the English language reader, it would be helpful to identify references that are only available in Chinese.

Reviewer 2 Report

Thank you for the opportunity to revise the paper “Identifying the predictors of community acceptance towards waste incineration plants in urban China: A Qualitative Analysis”. This paper focusses a very important political, economic, and social theme since community acceptance is crucial key-element for successful public policy implementation (Weible and Sabatier 2018). It is well developed and perfectly written, thus facilitating the reader understanding of the message. Even proposing acceptance for publishing in IJERPH there are a few things that I would like to discuss.

  • Theoretical approach and methodological options. Different approaches were used, i.e., social psychology, social governance, social exchange theory, among others, which makes it difficult to idealized data collection and even data treatment. It would be great to include a guide for interviews corresponding to each of these theories, namely, dimensions and questions;
  • Thematic analysis protocol. I miss a table of coding/recoding of themes and sub-themes since a Thematic analysis has been conducted. For example, were interviews variable-oriented? (Nowell et al., 2017)

Good luck with your research!

Reviewer 3 Report

         With the development of urbanization, cities absorb more and more people, and the problem of garbage disposal (building garbage dump) has become more and more important for social governance. In this context, the author uses a qualitative method to analyze the "contradiction" between residents and the government about the construction of waste treatment plants. In general, the topic selection has a certain practical significance, the use of qualitative analysis method also has certain characteristics, and the logical structure of the text is relatively reasonable. Some suggestions are as follows:

         (1) Information is asymmetric. In order to make an in-depth analysis of the public's acceptance of the construction of garbage dumps, it is not enough to only analyze the residents. The government and the main body of garbage plant construction should also make in-depth analysis. Common sense doesn't need to do the research to know that most people would not welcome a garbage plant near their home. However, the garbage plant is not built in this place, but should be built in another place. How to balance the relationship between the government, the main body of garbage plant construction and the residents is the top priority.

         (2) The representativeness of samples. In addition to the group expansion proposed in the first opinion, the author also needs to further describe the scope of the impact of the waste treatment plant (including the number of people, the scope of construction, the benefits after construction, etc.). As for the resident group alone, I still hold the opinion that I can get similar conclusions without doing such a study. At the same time, can the 22 groups represent all the groups within the scope of the waste treatment plant?

         (3) The result of qualitative analysis is confused with quantitative analysis. If rigorous statistical tests are conducted, we can conclude that X and Y are positively and significantly correlated. However, it is particularly strange to conclude that qualitative analysis can only be achieved by quantitative analysis. So some of the statements in the author's article seem strange.

Round 2

Reviewer 3 Report

The author has made careful modification according to my suggestion and recommend acceptance.